# A Study of the Essential Parameters of Friction-Stir Spot Welding That Affect the D/W Ratio of SSM6061 Aluminum Alloy

**DOI:** 10.3390/ma16010085

**Published:** 2022-12-22

**Authors:** Chaiyoot Meengam, Yongyuth Dunyakul, Siriporn Kuntongkum

**Affiliations:** 1Faculty of Industrial Technology, Songkhla Rajabhat University, Songkhla 90000, Thailand; 2Faculty of Engineering, Rajamangala University of Technology Srivijaya, Songkhla 90000, Thailand; 3Establishment of the Faculty of Engineering, Chaiyaphum Rajabhat University, Chaiyaphum 36000, Thailand

**Keywords:** friction-stir spot welding, D/W ratio, SSM 6061 aluminum alloy, tensile shear strength, globular structure

## Abstract

This study aimed to investigate how the depth-to-width (D/W) ratio of the welding area affects the welding quality of the SSM6061 aluminum alloy via the friction-stir spot welding (FSSW) process. The results showed that a higher D/W ratio directly results in better mechanical properties. If the D/W ratio value is high (at 1.494), then this leads to higher tensile shear strength at 2.25 kN. On the other hand, if the D/W ratio values are low (at 1.144), then this reduces tensile shear strength to 1.17 kN. The fracture surface behavior on the ring zone also affects the characteristics of ductile fracture. During Vickers hardness analysis, the hardness profiles are in the shape of a W; the maximum hardness was 71.97 HV, resulting from the rotation speed of 3500 rpm and the dwell time of 28 s, where the hardness of the base metal was at 67.18 HV. Finite element (FEM) analysis indicated that the maximum temperature during simulation was 467 °C in the region near the edge shoulder tool, which is 72.96% of the melting point. According to FEM simulation, the temperature under the tool pin region was 369 °C. The generated heat was sufficient to induce changes in the microstructure. For microstructure changes, the globular grain took on a rosette-like form, and coarse grains were observed in the thermal mechanical affect zone (TMAZ) and in the nugget zone (NZ), transforming in the mix zone. Hooks, kissing bonds, voids, and porosity are the defects found in this experiment. These defects indicate a discontinuity in the NZ that leads to worse mechanical properties. During examination via SEM and energy dispersive X-ray (EDX) analysis, the recrystallization structure from β-Mg_2_Si IMCs to Al_3_Mg_2_ and Al_12_Mg_17_ IMCs was observed. The size was reduced to an average width of 1–2 µm and an average length of 2–17 µm. Simultaneously, the oxides from the ambient atmosphere present during welding showed dominant partial elements from SiO_2_, MgO, and Al_2_O_3_.

## 1. Introduction

The AA6061 aluminum alloy is widely used in the automotive industry, aerospace industry, and railway industry [1]. The primary ingredients of the AA6061 aluminum alloy are magnesium and silicon, which are formed as Al-Mg-Si intermetallic compounds (IMCs). Its high tensile strength, excellent fatigue, hardness properties, and corrosion resistance make this alloy stand out [2]. Over the past several years, the microstructure and mechanical properties of the AA6061 aluminum alloy have been improved to achieve higher efficiency as a wrought aluminum format.

However, aluminum casting in a semi-solid state (SSM) is a new and improved technique [3]. J. Wannasin et al. [4] proposed that gas-induced semi-solid (GISS) is a modern, easy, and feasible production method because of its lower porosity, high-quality casting parts, and cost. Presented by R. Burapa et al. [5], GISS results in a fine and uniform Al-primary (α) globular structure that offers a better metal flow ability during casting, and ultimate tensile strength with excellent elongation were observed by S. Janudom et al. [6]. The ADC12 aluminum alloy can be easily applied to GISS casting, resulting in a uniform microstructure, no blistering, and little porosity. M. Honarmand et al. [7] investigated the additional Sr element content in the SSM A380 aluminum alloy when testing structural refinement to improve impact strength. M. Honarmand et al. found that the Sr-modified alloy also demonstrated a dominant globular structure morphology and resulted in a 40% increase in impact strength. Consequently, GISS aluminum alloys have begun to replace traditional casting methods, and the Al-Mg-Si alloy will be the first alloy group used in GISS casting due to its common use and excellent mechanical properties.

Friction stir spot welding (FSSW) is classified as solid-state welding and was invented by the TWI Institute [8]. FSSW development from the friction stir welding method (FSW) relies on a similar thermal generation mechanism. The bonded mechanism is produced by the heat input from the friction coefficient of the sample surface determined with a welding tool. The heat-input-generated thermal zone transforms the softening materials into an adhesion joint. The advantages of FSSW, such as non-filler metal, low power consumption, and high-strength joints, result in welded materials with different chemical properties and is environmentally friendly [9,10]. However, different essential parameters for FSSW often result from the changes in weld depth (D), weld width (W) in the nugget zone (NZ) [11,12], and microstructure recrystallization and mechanical properties [13]. For the FSSW of the previously studied AA6061 aluminum alloy in particular, Table 1 shows that the optimal parameters of FSSW used in the experiment that affect the joint efficiency [14]. Therefore, investigating the optimal parameters for FSSW on SSM6061 aluminum alloy is interesting to study.

The main concept of the experiments presented here is to investigate the effects of dependent parameters such as rotational speed and dwell time and constant parameters such as plunge depth, plunge rate, pin length, pin diameters, shoulder diameter, and diameter ratio on the FSSW of SSM6061 aluminum alloy plates. The depth-to-width (D/W) ratio in the NZ, tensile shear strength hardness properties, and heat input during welding simulated using finite elements (FE) were evaluated. After the FSSW of the SSM6061 aluminum alloys, the microstructure in the NZ, the thermal mechanical affect zone (AS-TMAZ and RS-TMAZ), and the defects on the base metal (BM) were evaluated and analyzed according to the phase transformation of Al-Mg-Si intermetallic compounds by means of optical microscopy (OM), scanning electron microscopy (SEM) and energy dispersive X-ray spectroscopy (EDX).

## 2. Materials and Methods

### 2.1. Materials

The material used for FSSW in this study was a 4 mm thick SSM6061 aluminum alloy plate. The process of producing semi-solid metal cast 6061 aluminum was as follows: The billet of wrought AA6061 aluminum alloy was melted to a semi-solid state at 605 °C. The nitrogen gas bubble flow rate was 19 L/min for 12 s through a graphite rod, and semi-solid materials were produced via the gas-induced semi-solid (GISS) technique discovered by Wannasin [19]. The average grain size of the SSM6061 aluminum alloy obtained from the α-primary aluminum matrix was 32–45 µm. The average grain size of the β-eutectic obtained from the intermetallic compounds of Mg_2_Si was 17–21 µm, with most grains being located in the grain boundary area shown in Figure 1. These resulted in very good weldability and very good corrosion resistance, and heat-treatability was observed [20]. The FSSW on the SSM6061 aluminum alloy used in this experiment was sponsored by GISSCO Company Limited (Songkhla, Thailand) [21]. The chemical composition and mechanical properties of the SSM6061 aluminum alloy are shown in Table 2.

### 2.2. Friction-Stir Spot Welding (FSSW) Process

For specimen preparation, the dimensions of each plate were 28 mm wide and 100 mm long. Then, two plates were connected to form a lap joint and were made to overlap by 28 mm. A specimen with a total length of 212 mm is shown in Figure 2. The welding tool was manufactured from SKH 57 high-strength steel and had a tool shoulder diameter of 20 mm and a pin diameter of 5.2 mm. The height of the pin was 3.2 mm and had a cylindrical straight shape. The calculation shows that the diameter ratio (D/d) was valued at 3.8, which is suitable for the relationship between the friction force and the thermal mechanism during welding, as shown in Figure 3.

FSSW experiments pay great attention to the essential parameters that affect the D/W ratio, microstructure, mechanical properties, and thermal mechanism. The results of previous studies have demonstrated that the tool rotation speed, dwell time, diameter ratio, plunge rate, plunge depth, and pin and shoulder size are significant for FSSW. Consequently, in this experiment, these essential parameters were determined to be necessary for successful welding at rotation speeds of 380, 760, 1240, 2700, and 3500 rpm (based on previous and preliminary studies); dwell times of 7, 14, 21, and 28 s; a constant plunge depth of 2.8 mm; a plunge rate of 18 mm/min; and the FSSW tool rotating in the clockwise direction. The essential parameters for the experiment are shown in Table 3. It is important that the FSSW is strongly clamped down because the rotational friction force can cause the specimen to fall off during testing. Therefore, the jigs and fixtures were designed to be strong enough for the specimen to be sufficiently clamped. Moreover, the bottom of the jigs and the fixture also prevent thermal dissipation through the use of insulating materials. Rotational speed and plunge rate control are part of an automatic system, with each step of the FSSW process being shown in Figure 4.

### 2.3. Finite Element Model (FEM)

The thermal changes that take place during FSSW are related to the D/W ratio, microstructure, and mechanical properties. Therefore, a thermal assessment was also conducted during FEM analysis. During FEM analysis, a personal computer (PC) with the following specifications: Intel^®^ Core^TM^, Intel Core i5-10300H central processing unit (CPU) 2.40 GHz, graphics processing unit (GPU) GeForce GTX 1650 Ti, and random access memory (RAM) Double Data Rate 4 (DDR4) 16 GB (Bangkok, Thailand; model: HP), was used to simulate the temperature distribution using Solidwork software with the COSMOS simulation module (Solidwork Ver. 2020). The FEM analysis was supported by Songkhla Rajabhat University, Songkhla, Thailand. The geometry of the specimen was created to be realistic by simulating the specimens to be assembled on top of each other in an overlapping joint orientation (Figure 5). The FEM carried out during COSMOS simulation consisted of three hexahedral elements, for a total of 63,369 and 91,730 nodes in the curvature-based mesh, and the size of the mesh ranged from 0.46 to 1.38 mm. During thermal simulation, transient thermal FEM was performed to observe the thermal distribution in the NZ. The thermal behavior demonstrated variation in the width and depth of the NZ; the thermal changes had a significant effect on tensile shear strength. In particular, the formation of a defect and specimen deformation was observed, which could represent another means through which the thermal changes result in a new crystallization phase on Al-Mg-Si intermetallic compounds.

### 2.4. Metallurgy and Mechanical Testing

After welding, the specimens for photography observation were used to measure the D/W ratio in the center of the weld using a light optical microscope from Olympus, the BX53M model, made in Tokyo, Japan. For microstructural studies, the specimen surface was prepared and polished with abrasive paper from TRUSCO (Tokyo, Japan). Finally, the specimen was etched using Keller regent for 60 s. A 190 mL amount of Keller regent was mixed in water with 5 mL of nitric acid; 3 mL of HNO_3_ was mixed in hydrochloric acid; and HCl was mixed with 2 mL in hydrofluoric acid. For the weld macrostructures, the plate was examined by means of scanning electron microscopy, and its dispersed elements were analyzed by an energy-dispersive X-ray spectrometer from FEI-Quanta, Zurich, Switzerland (model: 400FEG). For the Vickers hardness test, 100 g of force was applied to the indenter for 10 s, with a distance of 0.2 mm from the center, to obtain the hardness profiles. For the tensile shear strength, the investigator followed the procedures outlined by the American Society for Testing and Materials (ASTM) [22] using a Testometric testing machine (model FS500-AT) from Rochdale, UK. During the tensile shear strength test, the resulting stress–strain curve was recorded, and the average tensile shear strength was measured from three individual specimens with identical parameters. Eventually, the results for the tensile shear strength were taken to create the D/W ratio to measure the relationship to joint strength.

## 3. Results and Discussion

### 3.1. D/W Ratio

In the NZ, the tensile shear strength of the weld is influenced by the relationship of the depth with the width (D_NZ_/W_NZ_). The D/W ratio can be calculated from the width (from keyhole to the edge of the seam) of the NZ with the lap joint and the depth (from the surface to the middle of the workpiece) of the NZ around the keyhole, as shown in Figure 6. Therefore, the measurement area for the width and depth of the NZ is the region where globular grain changes were observed. The equation used to calculate the D/W ratio is shown below.
(1)D/Wratio=depthwidthofNZ

However, rotation speed, dwell time, and shoulder size are important mechanisms that generate heat [23], which is significantly related to changes in the D/W ratio. The heat generated during welding radiates to the lower sheet, completing the friction welding mechanism. When applied for the right amount of time, the successful attachment of two specimens was reported.

Table 4 shows how different D/W ratios of the SSM6061 aluminum alloy obtained by FSSW are compared to different experimental parameters. For 380 rpm, the D/W ratio is only 1.177–1.276, which is the lowest value obtained from this experiment. Notice that the D/W ratio for dwell times ranging from 7 s to 28 s relates to the thermal generation behavior and the friction between the surfaces of the shoulder with the surface of the material. Therefore, the rotational speed and dwell time are related to the D/W ratio. However, at the rotational speed of 1240 rpm and the dwell time of 14, more heat can spread more deeply. This results in a better D/W ratio. The highest optimal value of the D/W ratio was observed at 1.494. This is because the dwell time and rotation speed here are the appropriate conditions for thermal generation and are good enough to dissipate the thermal heat around the tool in when in a solid state. However, when the rotational speed was increased to the range of 2720 to 3500 rpm, the D/W ratio was significantly reduced. When the D/W ratio was in the range of 1.114–1.358, the transient thermal changes caused the material to be pushed away into flash, which affects the depth and width of the NZ [24]. Another reason to support the differences in the D/W ratio is different dwell times. A dwell time that is too long will result in a wider width, causing a lower D/W ratio. This also results in excessive thermal build-up behavior, leading to permanent thermal loss due to heat exchange behavior. Therefore, the higher the D/W ratio, the better the welding outcome.

### 3.2. Tensile Shear Strength and D/W Ratio Analysis

Tensile shear strength is directly related to the D/W ratio. Both the tensile shear strength or D/W ratio vary according to the rotation speed and dwell time. Rotation speeds and dwell times affect the relationship between the D/W ratio and tensile shear strength; the higher the D/W ratio, the better the tensile shear strength that can be observed. When the rotation speed was 1240 rpm and when the dwell time was 14 s, the maximum tensile shear strength was 2.25 kN. On the other hand, when the rotation speed was 3500 rpm and the dwell time was 28 s, the minimum tensile shear strength was 1.17 kN. Noticeably, a very long dwell time inevitably resulted in material loss around the weld. In other words, the dependence of tensile shear strength on dwell time in the D/W ratio was investigated. The D/W ratio obtained from the measurements and calculations is low, corresponding to a smaller joint area and low bonding strength. For a D/W ratio of 1.177, rotation speed of 380 rpm, and dwell time of 7 s, the tensile shear strength was 1.19 kN, in accordance with the rotation speed of 3500 rpm at the same dwell time, resulting in a tensile shear strength of 1.22 kN. The narrower joint area plays an important role in the low shear strength. Previous research on the joint areas affecting tensile strength has been carried out. This is further explained in Table 5, and the tensile shear strength results are consistent with previous research. The thermal generation for this Al-alloy group is significant because of the influences on tensile strength. The characteristics of the tear resulted in failure during the tension test, and most were born from hooks, which caused the initial crack during the tensile strength test, also observed by Silva et al. [25]. Consequently, thermal generation is associated with the experimental parameters, including the coefficients of friction (µ), thermal conductivity (q), angular velocity (ω), and axial force (P) [26]. Furthermore, another reason to explain the differences in tensile strength is the formation of defects contributing to the insufficient strength. The hook is considered as a defect that occurs near the NZ, resulting in lower weldability [27]. The crack often occurs in the NZ, where crack defects are caused by the inappropriate welding parameters, leading to bond strength failures, also reported by Zhou et al. [28]. Identically, porosity and voids are indicators of discontinuity in the NZ because of non-optimal parameters and inhomogeneity during recrystallization in the solid state [29]. The three types of defects mentioned above resulted in worse tensile shear strength. The tensile shear strength results obtained during the experiment are shown in Figure 7.

### 3.3. Characteristics of the Fracture Surface after Tensile Shear Strength Test

The shear fracture surfaces in the NZ of the lower sheet on the FSSW setup at a rotational speed of 1240 rpm and a dwell time of 14 s are shown in Figure 8. Fracture surfaces from different locations were evaluated using SEM micrographs with a top view of the surface, as shown in Figure 8a. Adhesion characteristics are caused by material formation on the upper sheet, and these formations occur around the keyholes and form a ring zone. It should be noted that the fracture locations have different characteristics according to structural and bonded strength. These locations are further described in Figure 8b–d. The fracture behavior is caused by loading, leading to permanent deformation and crack initialization, as shown in Figure 8b,d. The cracked surface is smooth and small in the direction perpendicular to the pull force. In this location, the grain boundary is separated, becoming a longer crack [37]. Simultaneously, the surface created by loading on the fractured surface is also enlarged, causing the cracks to expand in a semicircular path. This area shows ductile fractures, and the broken surface has a high level of roughness that can be clearly observed (Figure 8c). The final fracture locations can be observed in Figure 8d, with the material elongation in the pulling direction exhibiting elongated dimples. This overloading behavior and final transition into the plastic deformation failure stage are shown in Figure 8e [38]. The characteristics of the fracture surface shown in Figure 8c result in lower tensile shear strength. On the other hand, the characteristics of the fracture surface shown in Figure 8e cause higher tensile shear strength.

### 3.4. Microstructure of the FSSW of SSM6061 Aluminum Alloy

The macrostructures obtained at a rotational speed of 1240 rpm and a dwell time of 14 s can be observed in Figure 9. The NZ from both upper and lower sheets are different in terms of structural characteristics and the formation of defects. In the upper sheets, there is a wide weld area. On the other hand, the lower sheets show a narrower weld area in the NZ. That is because heat accumulation causes insufficient friction to be delivered underneath the specimen. This makes the NZ narrow, as reported by Fernandes et al. [39]. Changes in the base globular structure with a new crystallization structure are related to the present of defects, which can be observed in the upper sheets of AS-TMAZ, as shown in Figure 9a. These defects consist of micro voids in the NZ that are dispersed near the keyhole and that can be found among the longitudinal void around the seam between upper and lower sheets. This is caused by insufficient rotational speed or dwell time. Unlike the RS-TMAZ, in the upper sheets, more voids can be found on AS-TMAZ because the optimal temperature shows no evidence of causing shrinkage, but porosity was found. The low viscosity of the material was created during the thermal imperfection cycle (Figure 9b). However, in TMAZ, both sides are similar in terms of structural changes. The flow materials of α-primary aluminum formed as a laminar layer. The α-primary aluminum demonstrates elongated layer characteristics, in which the elongated direction corresponds to the clockwise direction along the welding direction. In addition, the NZ still has a fine homogenous combination of α-primary aluminum and β-Mg_2_Si that can be referred as the “Mix Zone”, as reported by Garg et al. [40]. For the mixed zone, the D/W ratio corresponding to tensile shear strength was investigated.

The microstructures of the RS-TMAZ on the lower sheets show the transformation from globular structures to uniform rosette-like structures under a hook. This incidence is obstructed by the joint boundaries for temperature distribution. Uncertainty and axial friction cannot react to each other, resulting in the incomplete crystallization of the rosette-like structures. The plunge rate and plunge depth parameters significantly influence the considered hook. This could have happened when the plunge rate and plunge depth increased, leading to the hook being larger in width. The retracting tool is another factor that widely affects the width of the hook during FSSW. There is thermal exchange between the surfaces of the materials and the tool’s shoulder. This is because the tooth tool affects the tool’s shoulder and possibly receives cold from the ambient atmosphere, alternating the attachment of the material to the shoulder and causing the hook to be wider, and this does not have a positive effect on the hook’s mechanical properties. Kissing bonds are another type of flaw evidenced in this study. Kissing bonds form around the end of a hook when the rotational speed is too high or too low, and the insertion of the oxide layer causing these kissing bond defects is shown in Figure 9c. Likewise, in the lower sheet of RS-TMAZ (Figure 9d), the presence of a hook is noticeable, and the hook is narrow compared to the lower sheets of AS-TMAZ because RS-TMAZ has good thermal generation. Consequently, the hook in RS-TMAZ was eliminated due to it being smaller in size. However, areas where the temperature is evenly distributed contribute to the development of globular structures on base materials, resulting in permanent changes. The thermal effect of globular grain expansion leads to grain growth, causing grains to form coarse globular characteristics [41]. The width of the weld is narrow and limited under the tool pin, as shown in Figure 9e,f. There is very little thermal energy in this area, causing the materials to not become soft, and this is coupled with the axial friction force, which cannot react at its full potential, leading to the risk of discontinuity. A void defect was in the area under the tool pin, as shown in Figure 9f. This experiment shows that the studies on how the process parameters of FSSW affect the thermal distribution in the NZ and TMAZ, which is related to the relationship between globular microstructures transformation and defect formation.

Figure 10 shows SEM micrographs of different characteristics of the Mg_2_Si IMCs from BM, AS-TMAZ, RS-TMAZ, and NZ, respectively. The results of the experiment clearly show the transformation of the Mg_2_Si IMCs into different shapes, sizes, and fragmentation types. The bases of the Mg_2_Si IMCs have interconnected shapes, such as gauze-like patterns. The average length of 33–57 µm and the average width of 2–11 µm are uniformly distributed. Wide distribution can be observed around the grain boundary in BM, as shown in Figure 10(Z1), where Z stands for ZONE. The very little thermal and friction is inaccessible to the BM; therefore, there is no change in the Mg_2_Si IMCs. While the Mg_2_Si IMCs in AS-TMAZ on bottom surface material lost their gauze-like shape, the gauze-like shape of the Mg_2_Si IMCs was broken by cyclic loading, resulting in a smaller size [42]. The average length was 22–31 µm, and the average width was 2–4 µm, with the elongation distribution being in accordance with the welding direction for Mg_2_Si IMCs shown in Figure 10(Z2). Likewise, particle size changes can also be observed in the Mg_2_Si IMCs in RS-TMAZ, with an average length of 16–23 µm and an average width of 2–3 µm. This cluster distribution is shown in Figure 10(Z3). Finally, the Mg_2_Si IMCs in the NZ showed the most changes: smaller grain size and an even distribution of Mg_2_Si IMCs, with an average width of 1–2 µm width and an average length of 2–17 µm, as shown in Figure 10(Z4).

The presence of atomic displacement and sliding Mg_2_Si IMCs is the cause of the microstructure changes. According to the differences in the shape of the Mg_2_Si IMCs at each location, the thermal cycle is an important contributor to the occurrence of this phenomenon. The change from Mg_2_Si IMCs to new IMCs under the limitation of thermal generation results in Al_3_Mg_2_ and Al_12_Mg_17_.

According to the EDX images, the percentage of composition near the NZ of the joined section was evaluated. Figure 11a shows the merger area of the NZ and TMAZ obtained by SEM photography. The amount of element concentration and the amount of intermixing between elements was mapped according to the color pattern shown in Figure 11b. The thermal energy generated during welding at the joint allowed the elements Mg, Si, and Al to move freely. The Mg, Si, and Al in the base formed Mg_2_Si due to the influence of thermal energy. It can be noted from the EDX mapping analysis that there were formations of aggregated compounds of Al_3_Mg_2_ in the NZ and of Al_12_Mg_17_ in TMAZ due to the constitution of the thermal mechanism, something that was also reported by Peng Chai et al. [43]. The intermetallic compounds Al_3_Mg_2_ and Al_12_Mg_17_ were formed in the solid state, but different sizes and shapes were observed. It is worth noting that the NZ and TMAZ were partially dominated by oxides, and it was found that SiO_2_, MgO, and Al_2_O_3_ were formed. During FSSW, oxidized Mg, Si, and Al were on the surface during periods of friction. The oxidization dragged down into the NZ. Subsequently, after welding was completed, the oxides were layered between the materials. However, SiO_2_, MgO, and Al_2_O_3_ formed as brittle structures [44]. The composition percentage for the EDX mapping analysis is shown in Table 6.

### 3.5. Finite Element Simulation for Temperature Distribution

A 3D model, created using finite element simulation, was used to study thermal generation and temperature distribution during the FSSW process. The optimal conditions (rotational speed of 1240 rpm, dwell time of 14 s) from previous studies were used to run the finite element analysis. The boundary condition was convective heat transfer of 210 W/m^2^K at the bottom of the lower sheets, whereas an equivalent conduction coefficient of 247 W/m^2^K was applied. The heat transfer on the entire surface of the specimen is called the “boundary condition”. The maximum temperature reached 467 °C at node number 22,468 in the region near the edge of the shoulder tool when the melting temperature of the SSM6061 aluminum alloy was about 640 °C. The calculations show that this maximum temperature is around 72.96% of the melting point of the base material. It is also recommended by Zhang et al. [23] that the optimum temperature must be in the range of 50–80% of the base material’s melting point. It is evident that in the region under the tool pin for the lower sheets, there is little thermal generation in the areas that cannot be reached by the thermal energy, which has a temperature of approximately 369 °C. Therefore, the appropriate temperatures for a successful FSSW process are not recommended to be lower than 369 °C, as indicated at node number 19,245. It is worth noting that there are higher thermal accumulations on RS-TMAZ, consistent with the hardness results for AS-TMAZ, which show high hardness properties. However, the thickness of the specimen affects temperature dissipation to the lower sheets. The ability of thermal energy to be distributed to thick specimens could be the cause of the imperfections in the weld. In other words, the thermal energy resulting from friction is a significant parameter to obtain successful FSSW joints. Consequently, in order to avoid some of the defects and discontinuities associated with solid-state welding [45], finite element simulation plays a role in investigations. Developed according to the results of many previous studies, Table 7 shows temperature data obtained by finite element simulation carried out using different software programs. The results of those studies were consistent with the results obtained in this research, especially regarding the thermal energy generated at the related shoulder and pin tool diameters. The results are dependent on the type of material, similar to what was found by Atak et al. [46]. The 3D model simulation analysis carried out using COSMOS finite element and Solidwork software (Version. 2020) is shown Figure 12.

### 3.6. Micro Vicker’s Hardness

The micro Vicker’s hardness profiles are shown in Figure 13. For a distinction in hardness that varies according to dwell time, the hardness distribution profiles are very similar for all conditions, with hardness profiles that are W-shaped or shaped like an upside-down M. Then, for the same area, the discontinuity of hardness in the region of the keyhole was formed by the tool pin. The hardness appears to be consistent with rotation speed and dwell time. This essential parameter influences the accumulation of thermal energy and the occurrence of permanent stress during plastic deformation. For a dwell time of 7 s and a rotation speed of 380 rpm, the hardness was about 60.85 HV. However, when increasing the rotation speed to 3500 rpm, the hardness also tends to be higher, about 65.52 HV, as shown in Figure 13a. However, this is still lower than the hardness of the base material at 67.18 HV. Certainly, Al_3_Mg_2_ and Al_12_Mg_17_ also contributed to the increase in hardness. The IMCs for Al_3_Mg_2_ and Al_12_Mg_17_ form small reinforcement particles [39]. Meanwhile, when the dwell time was increased, there was enough time for thermal energy to spread around the stirring tool, as seen in Figure 13b. At a dwell time of 14 s, a significantly higher average hardness was observed. The longer the dwell time, the higher the increase in hardness. However, if there is very high thermal build-up, this could result in the material over-softening, increasing the material flow to outside the ring zone and resulting in a flash defect. The results for a dwell time of 21 s clearly show a higher hardness value than the dwell times of 7 s and 14 s. This is because the basic principle of the FSSW process requires time to distribute the thermal energy properly. When the rotation speed reaches 3500 rpm, the hardness value is about 69.20 HV and about 59.33, 62.20, 61.00, and 64.07 HV for rotation speeds of 380, 760, 1240, and 2720 rpm, respectively (Figure 13c). Finally, the dynamic recrystallization in relatively high thermal conditions leads to fine-grain formation and exerts influence on hardness, which increases, as per Kittima Sillapasa et al. [51]. Likewise, the dwell time of 28 s and rotation speed of 3500 rpm offer very high hardness: 71.97 HV. However, high thermal energy during FSSW could contribute to the tool becoming worn out, which was also observed [52].

## 4. Conclusions

The above experiments were carried out to study how the essential parameters of the FSSW process affect the D/W ratio, microstructure, tensile shear strength, Vickers hardness, and thermal distribution of the aluminum alloy SSM6061. The conclusions obtained from this investigation are as follows:The D/W ratio values vary with essential parameters such as rotation speed and dwell time. Thus, the appropriate D/W ratio should be above 1.3. In the experiment, the calculated D/W ratio was 1.494, and a rotation speed of 1240 rpm and at a dwell time of 14 s made welding successful.The optimal tensile shear strength has a significant relationship with the D/W ratio value. The results show that that maximum tensile shear strength is 2.25 kN, related to a D/W ratio value of 1.494, whereas the tensile shear strength is less when the value of the D/W ratio is lower as well. The minimum tensile shear strength of 1.17 kN results in a D/W ratio of 1.144.After the tensile shear strength test, the fracture surface behavior on the ring zone of the lower sheets showed different fracture characteristics. Observed from the broken surface, the ductile fracturs the near ring zone are characterized by the differences in shallowness or deep fracture surface.Plastic deformation behavior occurs in the microstructure of the NZ and TMAZ. The NZ transformation between α-primary aluminum alloys with β-Mg_2_Si IMCs was also observed in the mixing zone. OM analysis showed that recrystallization changed from globular grain structures to uniform rosette-like structures. The coarse grain, the formation of a hook, kissing bonds, voids, and porosity were found in areas where there was insufficient thermal energy. From SEM analysis, surface morphology found that β-Mg_2_Si IMCs was broken into Al_3_Mg_2_ and Al_12_Mg_17_ IMCs, with an average width of 1–2 µm and an average length of 2–17 µm. For the EDX mapping evaluation at the NZ of the joint, the Mg, Si, and Al elements were able to move freely from the thermal cycle, and partial elements dominated by oxides to SiO_2_, MgO, and Al_2_O_3_ were observed.The finite element simulation analyzed the region near the edge shoulder tool at a maximum temperature of 467 °C or at 72.96% of the melting point of the base metal. However, the temperature of the region under the tool pin was 369 °C after being analyzed by COSMOS finite element analysis using Solidwork software.Due to the thermal changes, the hardness profiles were W-shaped or looked like an upside down M. The hardness tends to be higher at higher rotation speeds. The maximum hardness was 71.97 HV at a rotation speed of 3500 rpm and a dwell time of 28 s. Meanwhile, the Al_3_Mg_2_ and Al_12_Mg_17_ IMCs that are distributed in the NZ also contribute to increasing the hardness. Furthermore, stress from plastic deformation leads to hardness increases.

## Figures and Tables

**Figure 1 materials-16-00085-f001:**
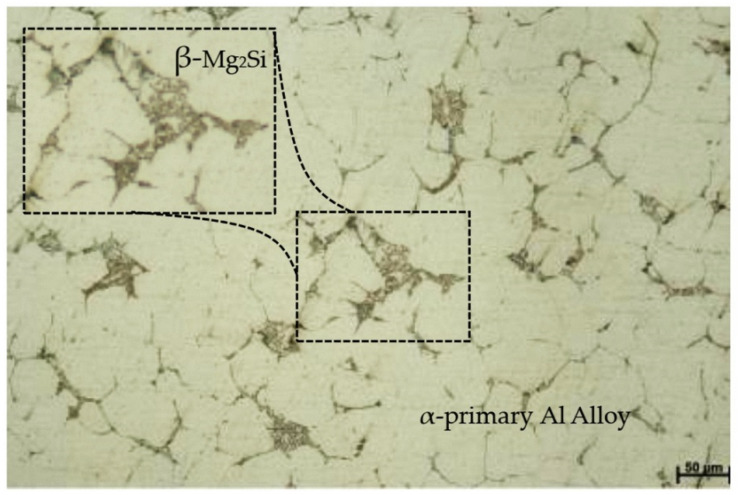
The photography of base globular microstructure on SSM6061 aluminum alloys.

**Figure 2 materials-16-00085-f002:**
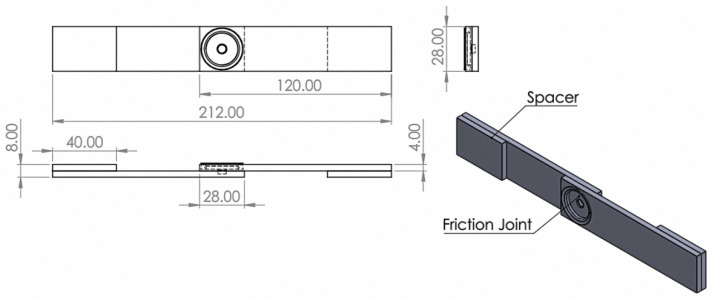
The geometry of the lap joint for the specimens of the experiment made using the FSSW process.

**Figure 3 materials-16-00085-f003:**
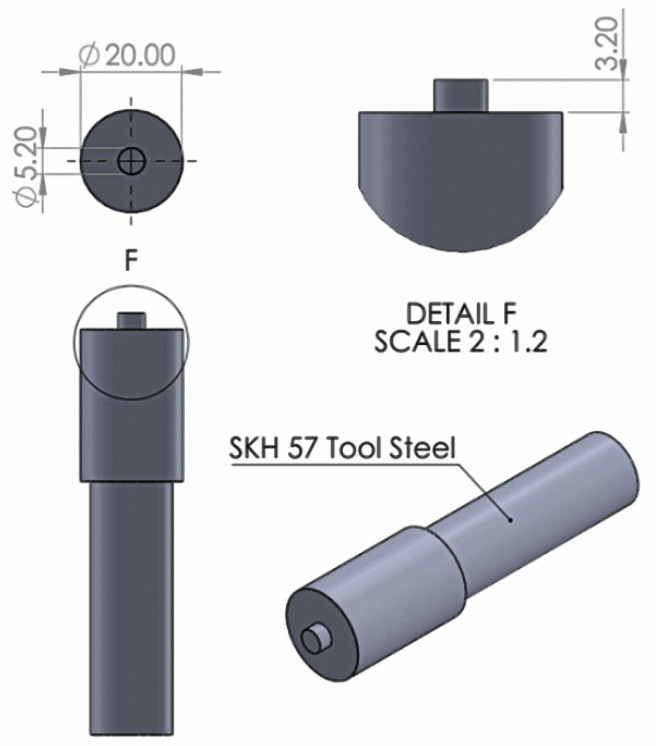
The geometry and dimensions of welding tool created using the FSSW process.

**Figure 4 materials-16-00085-f004:**
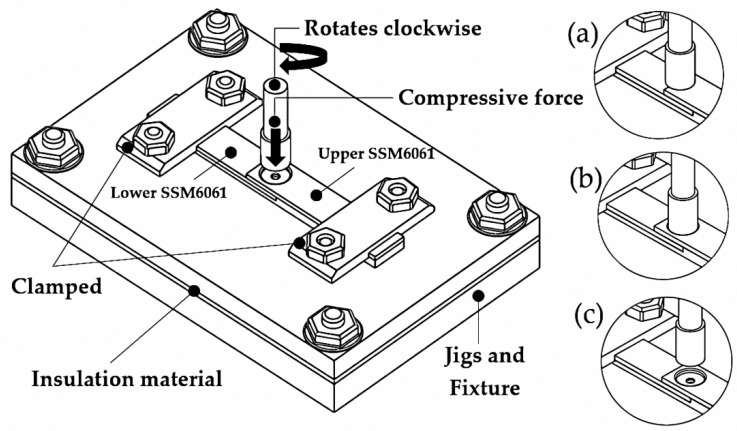
The schematic diagram steps of the FSSW process: (**a**) plunging, (**b**) stirring, and (**c**) retracting.

**Figure 5 materials-16-00085-f005:**
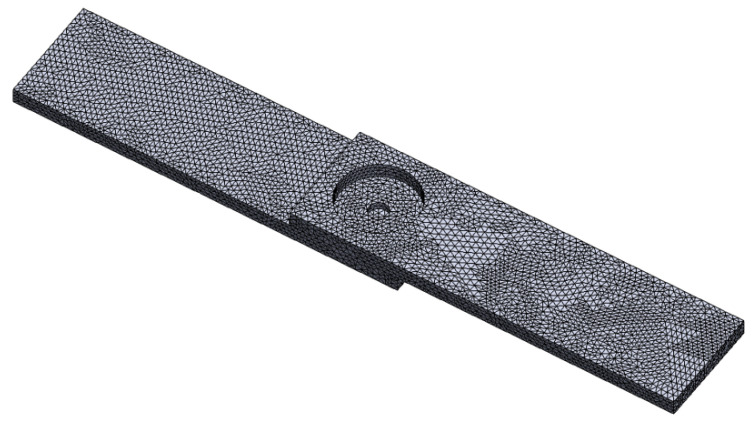
The contact conditions of the FEM and applied interface.

**Figure 6 materials-16-00085-f006:**
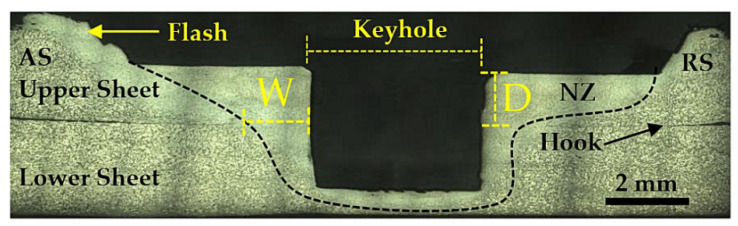
Micrographic of the cross-section of width and depth in NZ by FSSW of SSM6061 aluminum alloy.

**Figure 7 materials-16-00085-f007:**
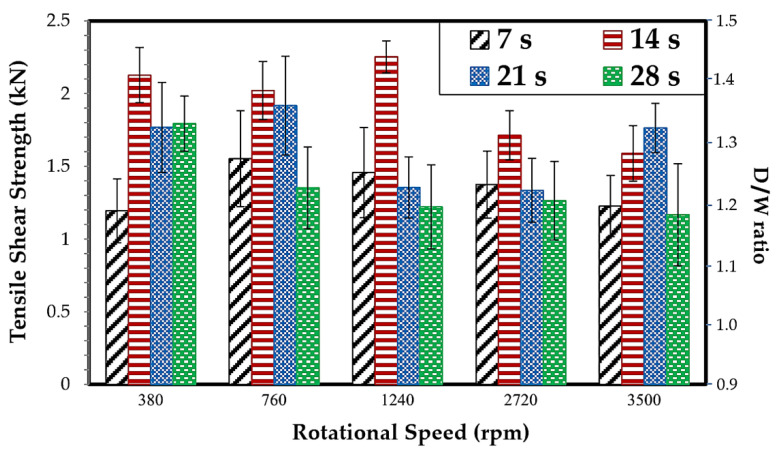
Diagram showing ensile shear strength relationships with D/W ratio by FSSW of the SSM6061 aluminum alloy.

**Figure 8 materials-16-00085-f008:**
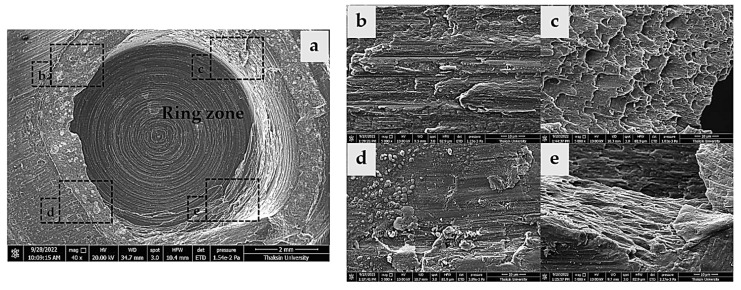
Characteristics of the failure fracture surface in the ring zone: (**a**) overall view of the shear fracture surface, (**b**) initial crack zone, (**c**) ductile fracture zone; (**d**) crack propagation zone; and (**e**) overload crack zone.

**Figure 9 materials-16-00085-f009:**
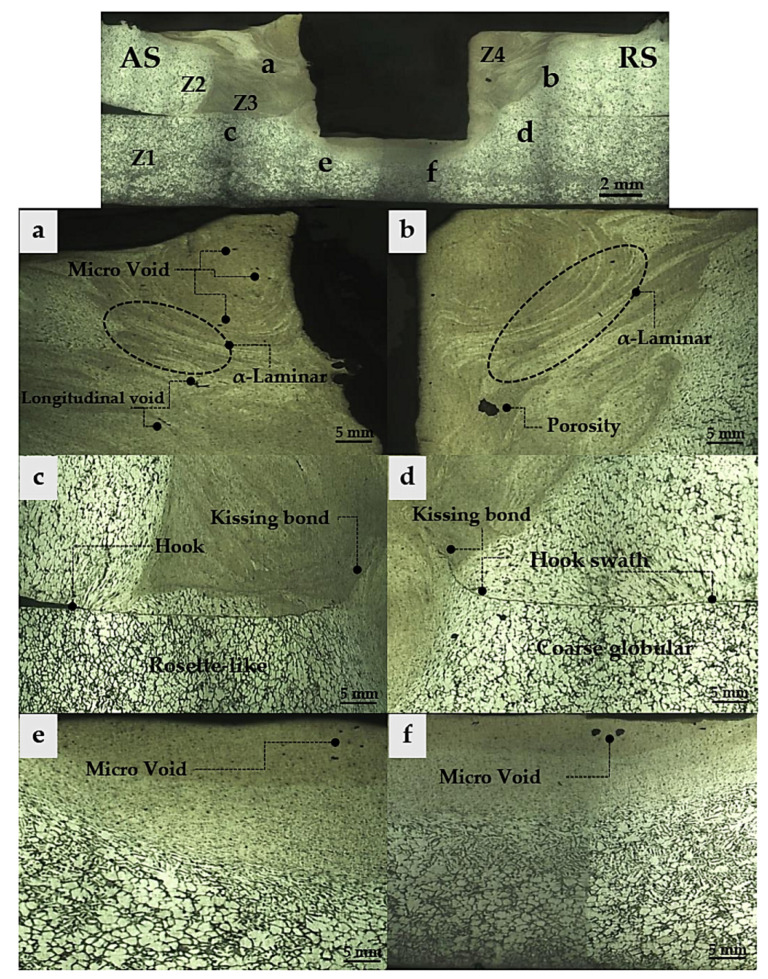
Micrographs of the cross-section of NZ from recrystallization by FSSW.

**Figure 10 materials-16-00085-f010:**
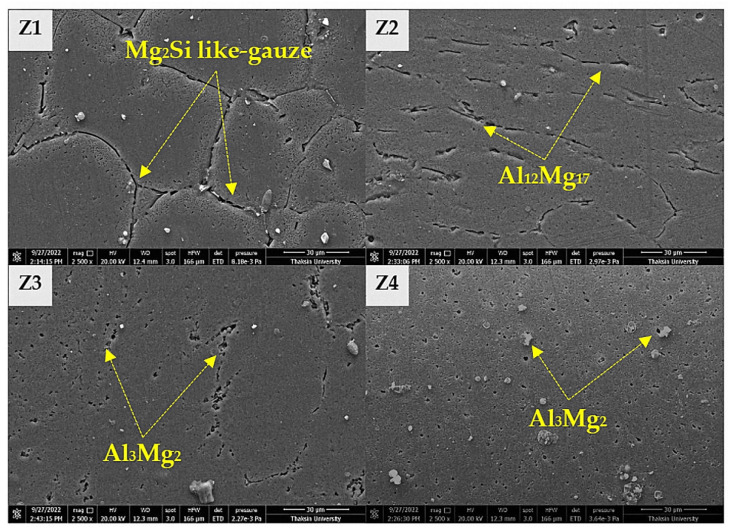
SEM micrographs of different characteristics of Mg_2_Si IMCs taken in EDX mode show the following: (**Z1**) BM, (**Z2**) RS-TMAZ, (**Z3**) AS-TMAZ, and (**Z4**) NZ.

**Figure 11 materials-16-00085-f011:**
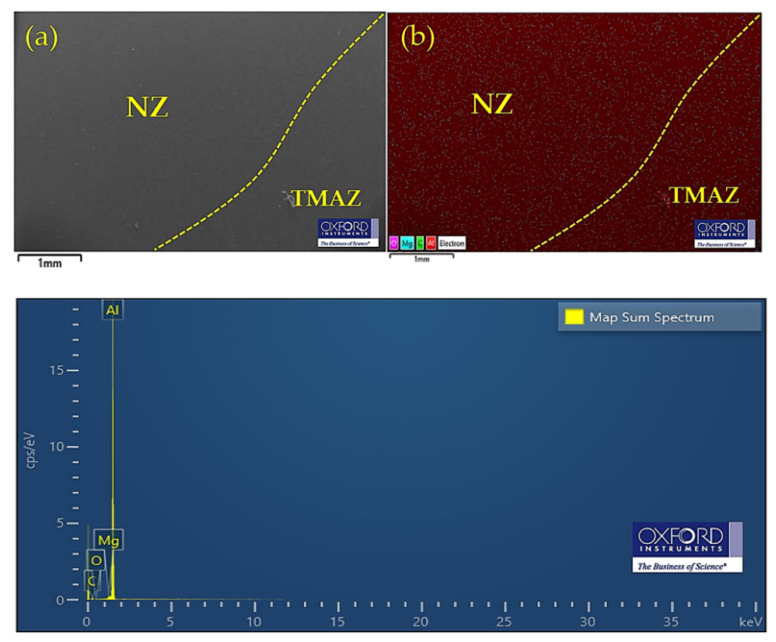
EDX mapping analysis of composition (wt%) in the NZ union of TMAZ: (**a**) SEM micrographs and (**b**) map showing color pattern analysis.

**Figure 12 materials-16-00085-f012:**
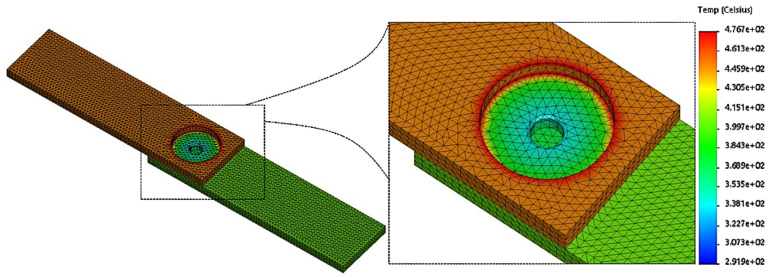
Temperature distribution in finite element simulation of SSM6061 aluminum alloy during FSSW process.

**Figure 13 materials-16-00085-f013:**
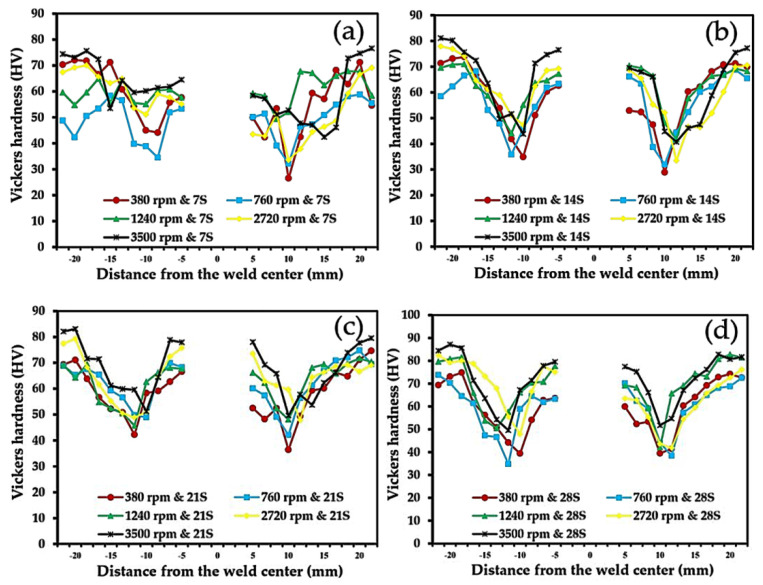
Micro Vickers hardness property of SSM 6061 aluminum alloy by FSSW according to dwell time: (**a**) 7 s, (**b**) 14 s, (**c**) 21 s, and (**d**) 28 s.

**Table 1 materials-16-00085-t001:** Highlights of previous research on the FSSW of 6061 aluminum alloys.

Workpiece	Pin Design	Shoulder Design	Diameter Ratio	Tool Materials	FSSW Parameters	Recommended Parameters	Reference
Material: AA6061-T6 Thickness: 2 mm	Shape: Round with M5 right-hand threaded Diameter: 5 mm Height: 2.85 mm	Shape: Flat Diameter: 12 mm	2.4	H13 tool steel	Tool speed: 10, 15, 20 and 25 mm/min Rotational speed: 1200, 1400, 1600, and 1800 rpm	Tool speed: 17 mm/min Rotational speed: 1387 rpm Lap shear load: 6134 N	[15]
Material: AA6061-T6 Thickness: 2.0 mm	Shape: Threaded cylindrical Diameter: 5 mm Height: 2.85 mm	Shape: Flat Diameter: 12 mm	2.4	H13 tool steel	Rotational speed: 900, 1200, 1400 and 1600 rpm Stirring time: 5 s Plunge depth: 0.2 mm	Rotational speed: 1400 rpm Tensile/shear strength: 5.31 KN	[16]
Material: AA 6061-T6 Thickness: 2 mm	Shape: Straight cylindrical Diameter: 8 mm	Shape: Flat Diameter: 18 mm	2.25	Not specified	Rotational speed: 900, 1120, 1400, and 1800 rpm Dwell time: 2 s Plunge depth: 0.3 mm	Rotational speed: 900 rpm Dwell time: 2 s Shear strength: 8416 N	[17]
Material: AA6061-O and T6 Thickness: 2 mm	Shape: Straight cylindrical Not pin	Shape: Flat Diameter: 10 mm	-	Not specified	Rotational speed: 1200 rpm Dwell time: 1.3 and 3 s Plunge depth: 0.9 mm Plunge rate: 20 mm/min	Rotational speed: 1200 rpm Dwell time: 1.3 s AA 6061-T6 Shear strength: 3261.7 N	[18]
Material: SSM6061 (As cast) Thickness: 4 mm	Shape: Straight cylindrical, Diameter: 5.2 mm Height: 3.2 mm	Shape: Flat Diameter: 20 mm	3.8	SKH57 steel	Rotational speed: 380, 760, 1240, 2720, and 3500 rpm Dwell time: 7, 14, 21, and 28 s Plunge depth: 2.8 mm Plunge rate: 18 mm/min	This research uses a new material for semi-solid-state casting. Of interest is the D/W ratio, thermal formation, microstructure, and mechanical properties, which have never been studied before	Present work

**Table 2 materials-16-00085-t002:** Chemical composition and mechanical properties of SSM6061 aluminum alloy.

Element (wt.%)	Si	Fe	Cu	Mn	Mg	Zn	Ti	Cr	Al
6061	0.80	0.70	0.40	0.15	1.20	0.25	0.15	0.35	Rem.
**Materials**	**Modulus of elasticity** **(GPa)**			**Yield strength** **(MPa)**		**Ultimate tensile** **strength (MPa)**			**Elongation** **(%)**
6061	68.9			276		310			12–17

**Table 3 materials-16-00085-t003:** Essential parameters and tool dimensions in FSSW experiment.

Parameters	Units	Values
Rotational speed	rpm	380, 760, 1240, 2720l, and 3500
Plunge rate	mm/min	18
Plunge depth	mm	2.8
Dwell time	s	7, 14, 21, and 28
Pin length	mm	3.2
Pin diameter	mm	5.2
Shoulder diameter	mm	20
Diameter ratio	---	3.8
Direction of FSSW	---	clockwise

**Table 4 materials-16-00085-t004:** D/W ratio of FSSW by SSM 6061 aluminum alloy.

Rotational Speed (rpm)	Dwell Time (s)	D (mm)	W (mm)	D/W Ratio
380	7	2.72	2.31	1.177
	14	3.18	2.41	1.319
	21	3.11	2.47	1.259
	28	2.82	2.21	1.276
760	7	2.84	2.36	1.203
	14	3.01	2.22	1.355
	21	3.07	2.34	1.311
	28	2.80	2.33	1.201
1240	7	2.99	2.34	1.277
	14	4.11	2.75	1.494
	21	2.87	2.31	1.242
	28	2.79	2.29	1.218
2720	7	2.91	2.33	1.248
	14	3.22	2.37	1.358
	21	3.03	2.35	1.289
	28	2.97	2.38	1.247
3500	7	2.66	2.30	1.156
	14	3.10	2.29	1.353
	21	3.51	2.55	1.376
	28	2.70	2.36	1.144

**Table 5 materials-16-00085-t005:** Detailed reports of tensile shear strength resulting from the FSSW process that have previously been studied.

Material	Optimal Parameters	Maximum Tensile-Shear Strength	References
AA3003-H12	1500 rpm/2 s	4.22 kN	[30]
AA2024-T3	950 rpm/15 s	6.90 kN	[31]
AA6022-T4/AM60B	1000 rpm/1 s	2.50 kN	[32]
Al-5083/St-12	900 rpm/14 s	4.02 kN	[33]
AA6061-T6	1800 rpm/20 s	9.52 kN	[34]
AA6061-T6	600 rpm/2 s	6.44 kN	[35]
AA6061-T4	2100 rpm/4 s	2.11 kN	[36]
SSM6061-As Cast	1240 rpm/14 s	2.25 kN	Present work

**Table 6 materials-16-00085-t006:** The composition percentage of NZ and TMAZ obtained from the FSSW process was measured via EDX mapping.

Element	Line Type	Apparent Concentration	k Ratio	wt%	wt% Sigma	Atomic %	Standard Label
C	K series	0.54	0.00539	41.73	0.75	60.94	C Vit
Si	K series	0.13	0.00043	2.55	0.24	2.8	SiO_2_
Mg	K series	0.1	0.00068	0.47	0.03	0.34	MgO
Al	K series	13.16	0.09453	55.25	0.72	35.92	Al_2_O_3_
Total:				100		100	

**Table 7 materials-16-00085-t007:** Summary of transient thermal characteristics of FSSW in previous investigations.

Material Study	Software	Temperature from Discovery	References
AA6061-T6	ABAQUS	514 °C	[47]
AA6060	ABAQUS	430–500 °C	[48]
AA2024-T3	ANSYS	316 °C	[49]
AA 7050/Ti64	LS-DYNA	400 °C	[50]
SSM6061-As Cast	COSMOS	467 °C	Present work

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
