# Peer review of "A Study of the Essential Parameters of Friction-Stir Spot Welding That Affect the D/W Ratio of SSM6061 Aluminum Alloy"

_materials, 2022, doi:10.3390/ma16010085_

Round 1

Reviewer 1 Report

Dear Authors: “The Study of Essential Parameters Friction Stir Spot Welding that affects from to D/W ratio of SSM6061 Aluminum Alloy”. I found the article interesting and well-prepared scientifically. Comments on the editorial page of the work, which in my opinion should be improved:

A.        Abstracct:

-            In my opinion, the summary should contain what the article has done, why and for what purpose. The abstract should be corrected.

B.      Introduction

-          The data presented in Table 1 from earlier works will be much more interesting if they are presented in the form of a description. What research has been done and what are the results. You can add what else is missing, thus justifying the presented work.

C.      Materials and Methods

-          Has the chemical composition of the material in Table 2 been tested? -          Fig. 1 is of very poor quality. Please correct if possible. -          In point 2.4 it would be good to add a photo of the microscope used for the research. D.      Results and Discussion -        Please explain why the authors state that: „However, at rotational speed of 1240 rpm and 14 s dwell time, the highest optimal value of D/W ratio was observed at 1.494.” I think this will improve the clarity of the article -        The drawings at work are of rather poor quality. And in Fig. 7, please enter the measurement error. -       Please comment on the results of numerical research and refer to their experimental verification. This is a very important issue.   In general, the research results and their analysis are interesting and prove a well-prepared work. E.       Conclusions In my opinion conclusions 5 and 6 should be improved. In their current form they should be taken into account in the analysis of research results, and they are not conclusions from them.

Author Response

 Comments and Suggestions for Authors (1)
(1) A. Abstract:
- In my opinion, the summary should contain what the article has
done, why and for what purpose. The abstract should be corrected.
Response 1: I edit the abstract section according to the comments from
the reviewer
(2) B. Introduction
- The data presented in Table 1 from earlier works will be much
more interesting if they are presented in the form of a description.
What research has been done and what are the results. You can add
what else is missing, thus justifying the presented work.
Response 2: I only put essential related factors that can link to the
current factors in my experiment into Table 1 only.
(3) C. Materials and Methods
- Has the chemical composition of the material in Table 2 been
tested? - Fig. 1 is of very poor quality. Please correct if possible.
- In point 2.4 it would be good to add a photo of the microscope
used for the research.
Response 3: Chemical compositions of the material were taken from
supplier.
I also improve the quality of figure one as suggested.
For point 2.4, I would like to omit not to put the picture of the
microscope because it is clearly stated the model of the microscope.

(4) D. Results and Discussion -
Please explain why the authors state that: „However, at rotational
speed of 1240 rpm and 14 s dwell time, the highest optimal value of
D/W ratio was observed at 1.494.” I think this will improve the
clarity of the article - The drawings at work are of rather poor
quality. And in Fig. 7, please enter the measurement error. - Please
comment on the results of numerical research and refer to their
experimental verification. This is a very important issue. In general,
the research results and their analysis are interesting and prove a
well-prepared work.
Response 4: I edit this sentence to “However, at rotational speed of
1240 rpm and 14 s dwell time, more heat can spread deeper. This will
cause better D/W ratio. The highest optimal value of D/W ratio was
obser
ved at 1.494.”
I improve the quality of the drawing. In figure 7, I add the error bar
into the chart.
I also add the comment on the result of numerical research and refer to
the experimental verification into a section as well.
(5) E. Conclusions
In my opinion conclusions 5 and 6 should be improved. In their
current form they should be taken into account in the analysis of
research results, and they are not conclusions from them.
Response 5: I believe that, for conclusion 5 and 6, the results were
already mentioned in the analysis part at section 3.5 and 3.6
respectively.

Reviewer 2 Report

The work is good and executed well. However following suggestions/corrections needs to be incorporated for publication.

1.      How you are selected these tool rotational speeds 380, 760, 1240, 2720 and 3500

2.      Please clarify why the D/W ratio fluctuates differently at the same speed and different dwell periods, especially D value is varying differently compared to W values.

3.      In Page 12, line 2 you are written like “The results of the experiment clearly show Mg2Si IMCs transformation in different shape, size, and fragmentation. The bases of Mg2Si IMCs have interconnected shapes like gauze.”  Need clarify and prrof of the above statement.

4.      Please improve the resolution of Fig. 13.

Why the hardness tends to be higher at a higher rotation speed.

Author Response

 Comments and Suggestions for Authors (2)
(1) How you are selected these tool rotational speeds 380, 760, 1240, 2720
and 3500
Response 1: I have selected this tool rotational speed based on the
previous studies and preliminary experiment. I also add this reason
into material and method section as well.
(2) Please clarify why the D/W ratio fluctuates differently at the same
speed and different dwell periods, especially D value is varying
differently compared to W values.
Response 2: the reason why D/W ratio fluctuates differently at the
same speed and different well period because the longer dwell time
would allow more heat to penetrate deeper to the workpiece.
However, if the dwell time is too long, the wider of heat could be
spread as well. Therefore, the right combination of rotational and
dwell time are the key the have the highest D/W ratio.
(3) In Page 12, line 2 you are written like “The results of the experiment
clearly show Mg2Si IMCs transformation in different shape, size, and
fragmentation. The bases of Mg
2Si IMCs have interconnected shapes
like gauze.” Need clarify and prrof of the above statement.
Response 3: I believe that, in Figure 10 (Z1), it is already shown in
yellow color that there is Mg
2Si like-gauze.
(4) Please improve the resolution of Fig. 13.
Why the hardness tends to be higher at a higher rotation speed.
Response 4: I improved the resolution of figure 13. The reason why
the hardness tend to be higher at the higher rotational speed is
because when the rotational speed is higher, the intermetallic
compound become more brittle and fine leading the better thermal
stress and better recrystallization were observed.

Reviewer 3 Report

High quality figures should be provided.

Author Response

 Comments and Suggestions for Authors (3)
(1) Summary
This work studied the friction stir spot welding for SSM6061 plates.
The rotation speed and time were optimized to achieve highest D/W
ratio, and tensile shear stress. Welding surface morphology,
deformation, defects and thermal distribution were characterized or
modeled.
Response 1: I edit the summary section according to the comments of
the authors
(2) General comments
Response 2: I believe that there is enough information for
introduction part. However, I modify all the pictures to a higher
resolution.
(3) Improvements that you could suggest on the paper
Major Improvement:
(
3.1) Page 2. The author mentioned about the “The bonded
mechanism is produced by the heat input……”. However,
there are some other mechanisms proposed. For example,
according to the recent study
(https://doi.org/10.1016/j.mtphys.2020.100252), “a nanoscale
amorphous layer can be introduced at the Al-Fe interface
without undesirable IMC.” More mechanism discussion
should be provided here.
Response 3.1: Thank you for your suggestion. I have looked at
the paper that the reviewer mentioned. However, it is not directly
related to this work.

(3.2) Page 2, Table 1. In this table, the author summarized previous
AA6061 studies. Has SSM6061 already been studied? Why not
include some SSM6061 results for comparison?
Response 3.2: I have not seen any SSM6061 produced by GasInduced Semi Solid (GISS) process. This work is the first one that I
try.
(3.3) Page 3. If some parameters are constant, how to study their
effect?
Response 3.3: the reason why the authors would like to put other
parameters into the chart because the authors would like to show the
possible factors that could inference the quality of the welding. Many
previous studies also only focus on dwell time and rotational speed
which are the major cause determining the weld quality. That’s why I
also focus on only these factors.
(3.4) Page 4, Figure 1. The author is suggested to provide higher
resolution images.
Response 3.4: I adjusted and improved all pictures to a higher
resolution.
(3.5) Page 4, section 2.2. What is the meaning of “D/W 3.8 is
suitable for the relationship between friction force and the
thermal mechanism”? It is a little hard to understand this
sentence.
Response 3.5: I believe that the reviewer misunderstood this
sentence. Actually, I would like to show the dimension of the tool pin
and tool shoulder ( D/d) ( not D/W). This dimension could generate
more heat if the D/d ratio is higher.

(3.6) Page 5, Figure 2,3. The author is suggested to provide higher
resolution image. They are very blurry.
Response 3.6: I improve the quality of all figures to a higher
resolution.
(3.7) Page 6, Table 3. So essentially, only rotation speed and dwell
time varied. How are other constant parameters, such as
plunge rate/depth, are determined? Why can they be treated
as optimal.
Response 3.7: after preliminary experiment we found that we
would like to fix plunge rate and depth because I believe that these
factors would not affect the weld quality much. Also, I would like to
minimize the number of tests in order to be able to analyze the other
factors.
(3.8) Page 7, section 3.1. The definition of D/W should be further
explained. The author mentioned
“maximum depth” and
“maximum width”. But from Figure 6, the “W” and “D” were
not obtained at the maximum width/depth location.
Response 3.8: I have changed this sentence to “The D/W ratio can
be calculated from the width
(from keyhole to the edge of the seem) of
the NZ with the lap joint and the depth
(from the surface to the middle
of the workpiece)
of the NZ around the keyhole shown in Figure 6.”.
(3.9) Page 8. “Therefore, the rotational speed and dwell time are
directly proportional to the D/W ratio.” This sentence
contradicts with the results in Table 4.
Response 3.9: I have rewritten this sentence to “Therefore, the
rotational speed and
dwell time related to the D/W ratio.”
(3.10) Page 8. “the occurrence of excessive thermal build-up
behavior resulting in permanent
thermal loss.” More
explanation should be provided on this. What is the meaning
of thermal build-up causes thermal loss. Why does thermal
loss cause lower D/W ratio.
Response 3.10: I edit this sentence to “Another reason to support
the differences of D/W ratio is different dwell time. If the dwell time is
too long, the wider width will present causing the lower D/W ratio. This
also provide excessive thermal build-up behavior resulting in
permanent thermal loss due to heat exchange behavior. Therefore, the
higher D/W ratio, the better welding outcome we can observed.”
(3.11) Page 9, Figure 7. Can we infer the 14s dwell time is always
optimal cross different speeds? If so, why?
Response 3.11: the reason why 14 seconds dwell time is always
optimal across different speed is because the longer dwell time could
generate heat loss causing incomplete recrystallization process which
directly affect the tensile strenght.
(3.12) Page 9. “the higher D/W ratio there was, the better tensile
shear strength there could be observed”. Why not display the
two results together, and analyze their correlation?
Response 3.12: Actually, Figure 7 has shown the D/W ration of
the right Y- axis. Therefore, this can explain the results in all aspects
completely.
(3.13) Page 10. It is not clear how is the facture surface related to
the tensile shear strength?
Response 3.13: I add the reason into the end of the paragraph: “If
the characteristics of the fracture surface shown in Figure 8(c) was

found, it causes lower tensile shear strength. On the other hand, if the
characteristics of the fracture surface shown in Figure 8(e) was found, it
causes higher tensile shear strength.”
(3.14) Page 13, Figure 10. How could SEM image determine the
IMCs, such as Mg
2Si?
Response 3.14: I run SEM with EDX mode. Therefore, it can
indicate the amount of intermetallic compounds.
(3.15) Page 14, Figure 11. “the elements intermixing was mapped
in color pattern shown in Figure.11(b).” It is almost
impossible to see the dots clearly.
Response 3.15: I improve the image into a bigger size and higher
resolution.
(3.16) Page 14, section 3.5. “The maximum temperature reaches to
467 °C”. Under what condition? Speed and time?
Response 3.16: I add more detail in the second sentence into this
section
: “The optimal condition (rotational speed at 1240 rpm, 14s dwell
time) from previous study were used to run
in Finite Element.”
(3.17) Page 16, Figure 13. Why not taking a sample point at the
center?
Response 3.17: the reason why I didn't take the sample point at
the center because it is the keyhole area which does not allow to be
measured from the Vickers hardness machine I used.

Round 2

Reviewer 3 Report

Thanks the author for addressing all comments properly.